# Relay Selection for Over-the-Air Computation Achieving Both Long Lifetime and High Reliability

**DOI:** 10.3390/s23083824

**Published:** 2023-04-08

**Authors:** Jingyang Zhou, Suhua Tang

**Affiliations:** Department of Computer and Network Engineering, The University of Electro-Communications, Tokyo 182-8585, Japan

**Keywords:** Internet of Things, over-the-air computation (AirComp), relay selection

## Abstract

In a general wireless sensor network, a sink node collects data from each node successively and then post-processes the data to obtain useful information. However, conventional methods have a scalability problem: the data collection/processing time increases with the number of nodes, and frequent transmission collisions degrade spectrum efficiency. If only statistical values of the data are needed, using over-the-air computation (AirComp) can efficiently perform data collection and computation. However, AirComp also has its problems: when the channel gain of a node is too low, (i) the transmission power of that node will be high, decreasing the lifetime of that node and the entire network, and (ii) sometimes, the computation error still occurs even though the maximal transmission power is used. To jointly solve these two problems, in this paper we investigate the relay communication for AirComp and study a relay selection protocol. The basic method selects an ordinary node with a good channel condition as a relay node, considering both computation error and power consumption. This method is further enhanced to explicitly consider network lifetime in relay selection. Extensive simulation evaluations confirm that the proposed method helps to prolong the lifetime of the entire network and reduce computation errors as well.

## 1. Introduction

With the wide-spread of the Internet of Things, there will be more and more context-aware applications that collect environmental information through sensor nodes (hereinafter referred to as nodes) and make corresponding actions in real time in the future smart society. Nodes will be deployed and connected to the Internet through technologies such as NB-IoT and LoRa [1]. Generally, during the data collection process, the sink node (hereinafter referred to as sink) needs to receive data from each node one by one. When there are tens of thousands of nodes within the coverage of a sink, it will take a lot of time to receive all the data. In addition, the nodes share the same channel in the communication. If the network uses the CSMA (carrier-sense multiple access) protocol for channel access, the increase in the number of nodes will lead to frequent transmission collisions, which further degrades spectrum efficiency. 

In some IoT applications, people only want the statistics of the data (e.g., the average temperature or the average noise level in a certain area and so on), not the individual data collected by nodes. For such applications, there is an efficient method called over-the-air computation (AirComp) [2,3]. In AirComp, nodes first preprocess the data, and then all nodes simultaneously transmit the preprocessed data through analog signals [4]. All signals add together at the antenna of the sink, which corresponds to the sum operation. By using analog AirComp, the computation error can be made smaller than that of digital schemes when using the same amount of resources (power and bandwidth) [5]. As well as the sum operation, AirComp also can support any kind of nomographic functions by proper preprocessing and postprocessing [6,7,8]. Recently, deep AirComp has been studied, using federated learning in the pre-processing and post-processing stages, which enables more advanced processing of sensor data [9].

Typically, in order to ensure unbiased data fusion, all nodes use transmission power control [10,11] to achieve a consistent signal magnitude when all signals arrive at the sink. Two problems may occur in the original AirComp model:For nodes far away from the sink with low channel gain, greater transmission power is required to reach the target magnitude in order to minimize the computation error. This will result in a short lifetime of nodes and the entire network.Under the transmission power limit, some nodes cannot reach the target magnitude even though they use the maximum transmission power. This leads to the computation error, which is measured by mean squared error (MSE).

Various methods have been studied to reduce the MSE from different aspects. The basic method is to apply transmission power control [10,11] using high power when the channel gain is low. However, this alone cannot well solve the problem when some nodes are in deep fading or far from the sink.

Conventional methods to channel fading can be applied in AirComp, but simultaneous transmission of multiple signals needs to be taken into account carefully. Time diversity is considered in [12], where the computation is distributed to multiple slots, and each node can independently select one slot to transmit its signal based on its channel gain. The optimal number of slots is formulated, considering the benefit of channel gain improvement and the demerit of increased noise power due to using multiple slots. A counterpart in the frequency domain is studied in [13], where the computation is distributed over multiple channels.

Another effective method to channel fading is path diversity, e.g., relay for network coding [14]. Because AirComp usually works for analog signals, amplify-and-forward based relay is investigated in [15] using a dedicated relay node to help nodes with low channel gains. Two relay policies, Simple Relay Policy (SRP) and Coherent Relay Policy (CRP), are studied. The two relay policies are further refined with a new metric, by which the transmission power is reduced while the MSE is kept low. However, the transmission power of the relay node may be much higher than that of normal nodes, leading to a surge in power consumption. In addition, the impact of the relay node on the network lifetime is not examined in this article. Under the constraint of transmission power, not all nodes can use the relay. In [16], node scheduling, deciding which nodes can use the relay, is studied. Recently, new mechanisms, such as intelligent reflective surfaces, were studied in [17]. The phases of reflective items can be adjusted so that signals add constructively at the sink. However, signals are not amplified at the reflective surface, and usually many reflective items are required to reach a reasonable signal strength. In addition, when multiple signals use the same reflective surface, the control of the surface phase is a non-convex problem.

It is also possible to use multiple antennas at the sink [18]. However, the alignment in channel gain is different from beamforming [19] because in AirComp different signals add together, not necessarily enhancing each other. The control of the antenna array is a non-convex problem, and it is difficult to find the optimal solution. Extending the distance between antennas, simultaneous receiving at multiple sinks is studied in [20], where the interference needs to be taken into account.

The knowledge of channel gains at the sink and the synchronization between nodes and the sink usually are assumed, which, however, is non-trivial. Channel estimation is studied in [18], using an iterative method with multiple antennas. The statistics of channel gains instead of their instantaneous values are exploited in [12]. A more aggressive policy, blind AirComp, is explored in [21].

Typically, it is assumed that signals from all sensors are non-correlated, which simplifies the analysis. However, when sensors are densely deployed, or in some specific applications (e.g., when AirComp is used for model aggregation in federated learning), signals from multiple nodes are correlated. This problem is explored in [22,23], which requires the correlation information and is not well solved yet.

As for power consumption, in [24], Zang et al. try to minimize the sum power of the sensors under the constraint of MSE. The results show that the sensors with poor or good channel conditions should use less power than the ones with moderate channel conditions to reduce the MSE and the sum power of nodes. However, the random variation of channel gains is not taken into account.

To solve the aforementioned problems, in this paper, we investigate relay communication for AirComp, and use relay selection to both extend network lifetime and reduce MSE. To the best of our knowledge, this is the first work on relay selection for AirComp. In this method, each node and the sink are equipped with a single antenna. It is assumed that signals from all nodes are non-correlated and channel gains are known at the sink. We first calculate a set of candidate relay nodes based on channel gains. Then, an evaluation metric, considering transmission power, remaining battery capacity, and MSE, is computed for each candidate, based on which the optimal relay is selected. In addition, the relay node is periodically selected at regular intervals to avoid over-consumption of power at the relay node. This basic method is further enhanced, explicitly considering the network lifetime in the relay selection. Extensive simulation evaluations show that the basic method can extend the network lifetime and reduce MSE, while the enhanced method can further prolong the network lifetime under the same MSE.

In the rest of this paper, Section 2 reviews the related work on the AirComp model, especially the relay transmission methods. Then, Section 3 presents the proposed method, both the selection of candidate relay nodes and the metric for evaluating the suitability of candidate relay nodes. Section 4 describes the simulation setting and analyzes the results and factors affecting the lifetime and MSE. Finally, Section 5 concludes this paper and points out future works.

## 2. Related Research

### 2.1. Over-the-Air Computation [10]

A sensor network consisting of K nodes and one sink is considered, as shown in Figure 1. Node k synchronizes its time with the sink using the control signal from the sink, preprocesses its data (sk) by ψ(·) and transmits its data as an analog signal (xk∈ℂ, |xk| ≤ v, E{|xk|2}=1) at the same time together with other nodes. These signals are summed up at the sink antenna, which is postprocessed by ϕ(·). For simplicity, here, the sum operation is considered: ψ(t)=t is used in the preprocessing and ϕ(t)=at is used in the postprocessing, where a is the Rx-scaling parameter. In order to eliminate the difference in channel coefficient (hk) among nodes, the transmission power is adjusted by a Tx-scaling factor bk and the magnitudes (hkbk) of all signals are aligned to the same value if the required power is less than the maximum. The received signal at the sink is
(1)r=a ·(∑k=1Khkbkxk+n),
where *n* is additive white Gaussian noise (AWGN) with mean value 0 and variance σ2. a, hk, and bk are complex numbers. Because it is possible to adjust the phase of bk to ensure that hkbk is positive real, in the analysis, usually it is assumed that a, bk, and hk are positive real for simplicity [10]. In the evaluation, complex numbers are used instead.

Assume all signals are non-correlated and independent of the noise. The mean squared error ($\mathrm{MSE}\;=\;E\left\{{\left|r-\overset{K}{\underset{k=1}{\sum }}x_{k}\right|}^{2}\right\}$), between the received signal r and the target result $\overset{K}{\underset{k=1}{\sum }}x_{k}$, is
(2)$$\mathrm{MSE}\;=\overset{K}{\underset{k=1}{\sum }}|ah_{k}b_{k}-1{|}^{2}+|a{|}^{2}\sigma^{2},$$
where $P_{k}=b_{k}^{2}\;\leq \;P_{max}$. This MSE may be caused by channel fading and noise and is jointly determined by a and $b_{k}$. 

As shown in Figure 2, when the channel coefficients $h_{k}$ of some nodes are too low, the target magnitude cannot be reached even with the maximum transmission power. Then, MSE occurs in the received signal.

### 2.2. Non-Relay Methods for Reducing Power Consumption

In [24], Zang et al. transform the non-convex problem in Equation (2) into a convex problem by setting ${\hat{b}}_{k}=ab_{k}$ as follows:(3)$$\underset{\left\{{\hat{b}}_{k}\right\}}{\min}\;\mathrm{MSE}\;=\sum_{k=1}^{K}|h_{k}{\hat{b}}_{k}-1{|}^{2}+\frac{\sigma^{2}}{P_{max}}\sum_{k=1}^{K}{\left|{\hat{b}}_{k}\right|}^{2}.$$Then, they minimize the sum power of the sensors under the constraint of MSE. The results in [24] show that the sensors with poor or good channel conditions should use less power than the ones with moderate channel conditions to reduce the MSE and the sum power of nodes. However, they did not consider random channel gains ($h_{k}$) but set the channel gains of all the nodes in the simulation to be a series of equal differences, which is not practical.

### 2.3. Relay Methods

The amplify-and-forward-based relay model in [15] is shown in Figure 3. In addition to nodes and sink *d*, a dedicated relay node *r* is added. The nodes are divided into two groups ($N_{r}$ and ${\bar{N}}_{r}$) according to channel gains between nodes and the relay node *r*/sink *d*. A general policy is that nodes ($k\in N_{r}$), far from the sink but close to the relay, use the relay node *r*, and nodes ($k\in {\bar{N}}_{r}$) close to the sink send directly to the sink. 

The communication is performed in two slots. bk,1,bk,2∈ℂ reflect the transmission power of node k in two slots. nr,1 and nd,2 are the AWGN at *r* in the first slot and at d in the second slot, respectively. ar,1,ad,2∈ℂ are the receive factors at r and d, respectively.

In the first slot, nodes using relay ($k\in N_{r}$) transmit their signals to relay node *r*. The signal received at relay *r* is
(4)$$s_{r,1}=a_{r,1}\cdot (\sum_{k\in N_{r}}h_{k,r}b_{k,1}x_{k}+n_{r,1}).$$

Because the relay node helps forward multiple signals in the second slot, it tends to consume a lot of power. The transmission power of the relay node is
(5)$$P_{r,\mathrm{TX}}={\left|\frac{1}{a_{d,2}h_{r,d}}\right|}^{2}\cdot \underset{k\in N_{r}}{\sum }{\left|a_{r,1}h_{k,r}b_{k,1}\right|}^{2}.$$

Based on the difference of communication nodes in the second slot, the relay model can be divided into two types: SRP and CRP.

#### 2.3.1. Relay Communication Policy—SRP

In SRP, in the second slot, nodes not using relay (k∈N¯r) transmit their signals to the sink and the relay node forwards its received signal to the sink. The signal received at the sink is
(6)sd,2=ad,2·(∑k∈N¯r ∪{r} hk,dbk,2xk+nd,2).
Each node only transmits in one slot. For k∈Nr, bk,1∈ℂ, bk,2=0. For k∈N¯r, bk,1 =0, bk,2∈ℂ. 

#### 2.3.2. Relay Communication Policy—CRP

The CRP communication method is shown on the right side of Figure 3. In the second slot, the CRP allows all nodes to transmit signals to the sink, not just k∈N¯r and the relay node. Then, the signal received at the sink is
(7)sd,2=ad,2 ·(∑k∈Nr∪N¯r∪{r} hk,dbk,2xk+nd,2).For k∈Nr, each node transmits its signal in two slots, and there is an overall power constraint per node, |bk,1|2+|bk,2|2≤Pmax. For k∈N¯r, bk,1=0, bk,2∈ℂ.

From nodes $k\in N_{r}$, the sink receives two copies of the same signal in the second slot: one with a magnitude $A_{k,r}$, forwarded by the relay, and the other with a magnitude $A_{k,d}$, sent directly by a node itself. With proper control of the Tx-scale phases, the two copies of the same signal add coherently at the sink, and the overall magnitude is
(8)$$A_{k}=A_{k,r}+A_{k,d}.$$

With the coherent combination of two copies of the same signal, in CRP, at the first slot, nodes using relay use a part of their power to transmit their signals to the relay node, and the relay node does not need to forward the signal with full magnitude in the second slot. This helps to reduce the transmission power of the relay node.

### 2.4. The Problems

The above relay methods can effectively reduce the MSE of AirComp through relay transmission. However, there are still two problems:The relay node is a dedicated node without power constraint, which increases the system cost.Reducing MSE usually is achieved at the cost of increased transmission power at the nodes, which decreases the network lifetime.

## 3. Proposed Method

In this paper, instead of using a dedicated relay node, we propose a new method to properly select an ordinary node as a relay node in the AirComp network. This method not only considers the power consumption and maximum power constraint of nodes in the network but also reduces the MSE as much as possible. Therefore, we call it relay Selection considering both Lifetime And MSE (SLAM). Here, network lifetime is defined as the difference between the beginning of communications in the network and the time when the first node in the network depletes its energy.

It is already proven that CRP works better than SRP in the amplify-and-forward-based relay [15]. Therefore, in the following we mainly study how to dynamically select a relay for CRP.

### 3.1. System Model

All nodes and the sink use a single antenna. Nodes synchronize their time with the sink. The sink knows the channel coefficient (h) between itself and all nodes and the channel coefficients among all nodes, and performs centralized relay selection. The estimation of channel gain is left for future work. Block fading is assumed, i.e., the channel coefficient is stable within two slots.

Because AirComp is performed on analog signals, amplify-and-forward (AF)-based relay is used, in the same way as in [15]. However, different from the dedicated relay node used in [15], we chose an ordinary node in the network as a relay node. This choice also caused us to pay attention to the transmission power of the relay node to avoid an over-increase in its power consumption. Because the relay node has to transmit its own signal as well, its transmission power is
(9)$$P_{r,\mathrm{TX}}={\left|\frac{1}{a_{d,2}h_{r,d}}\right|}^{2}\cdot \left(1+\underset{k\in N_{r}}{\sum }{\left|a_{r,1}h_{k,r}b_{k,1}\right|}^{2}\right).$$

The transmission power of the relay node is decided by the nodes using the relay, under the maximal power constraint. Therefore, it always approaches, but is less than, the maximal power constraint.

The proposed method can be roughly divided into two modules—the calculation of candidate relay nodes and the selection of a relay node from the candidates, as shown in Figure 4.

### 3.2. Calculation of Candidate Relay Nodes

Two conditions are given for deciding whether a node can be used as a relay node for over-the-air computation, as follows:The transmission power of a node using the relay is less than that in the basic AirComp model, i.e., the relay node should help to reduce the transmission power of other nodes using relay.The relay node’s transmission power is less than the maximum value.

We designed Algorithm 1 to select candidate relay nodes. In Algorithm 1, we regarded each ordinary node in turn as a relay node. When regarding node i as a relay node (Line 5), we computed the difference of channel gains to relay and sink as D (Line 7) and sorted nodes in an ascending order of D (Line 8). This is to let nodes far away from the sink but near to the relay use the relay node first. Then, with each possible number of nodes that can use node i as the relay, we recorded the corresponding MSE (Line 9–13). Finally, we found the optimal number of nodes using relay according to the MSE value, and recorded nodes’ power and MSE (Line 14). After all nodes were iterated, we sorted ${\mathrm{MSE}}_{i}$ in an ascending order (Line 16) and selected a number of top nodes as relay candidates R (Line 17).
**Algorithm 1:** Select candidate relay nodes**Procedure**: Find candidate relay nodes**Input**: $h_{k,d}$: Channel coefficient between sink and node, $\;\;\;\;\;\;\;\;\;\;\;h_{i,j}$: Channel coefficient between nodes i and j;**Output**: R: A set of candidate relay nodes;**For** i=1,2,⋯,K **do**|  Regard node i as a relay node r|  Get $h_{k,r}$ (channel coefficient between relay r and node k);|  Calculate the difference of channel gain to relay and sink, $D=\{|h_{k,d}\left|-|h_{k,r}\right|\}$;|  Sort D in the ascending order;|  **For** j=1,2,⋯,K,j≠i **do**|  |  Assume top *j* nodes in D use the relay node;|  |  Minimize MSE of CRP, with power constraint for all nodes including relay;|  |  Calculate transmission power ($P_{j}^{i}$) of all nodes;|  **End**|  Find optimal number of nodes that can use node i as relay and record ${\mathrm{MSE}}_{i}$**End**Sort ${\mathrm{MSE}}_{i}$ in the ascending order;Select a number of top nodes as relay candidates R based on ${\mathrm{MSE}}_{i}$;**Return** R

### 3.3. Relay Selection Considering Both MSE and Remaining Energy

Here, we present the evaluation metric for selecting a node from the candidate relay nodes as the relay.

#### 3.3.1. Evaluation Metric for Selecting Relay

Relay selection should consider both MSE and the remaining energy of candidate relay nodes. Usually, it is preferred to select for the relay a node that both has a large remaining energy (long lifetime) and a small MSE. To this end, we designed a new metric to evaluate the suitability of each candidate as a relay, considering both factors, as follows:(10)$$\theta_{i}=\alpha \cdot w_{E,i}+\left(1-\alpha \right)\cdot w_{MSE,i},\;w_{E,i}=\frac{E_{i}-\underset{i\in R}{\min}E_{i}}{\underset{i\in R}{\max}E_{i}-\underset{i\in R}{\min}E_{i}},w_{\mathrm{MSE},i}=-\frac{{\mathrm{MSE}}_{i}-\underset{i\in R}{\min}{\mathrm{MSE}}_{i}}{\underset{i\in R}{\max}{\mathrm{MSE}}_{i}-\underset{i\in R}{\min}{\mathrm{MSE}}_{i}},$$
where $E_{i}$ is the remaining energy of node i, $w_{E,i}$ is the normalized value of $E_{i}$, $w_{\mathrm{MSE},i}$ is the normalized value of ${\mathrm{MSE}}_{i}$, and α and 1−α denote the weights of remaining energy and MSE, respectively. Here, we multiply the value of MSE by −1 in the evaluation metric to ensure that for both $w_{E,i}$ and $w_{\mathrm{MSE},i}$, a large value means a better suitability.

Using Equation (10), we can compute the metric for each candidate relay node in R. Then, from R, the node with the largest metric $\theta_{i}$ is selected as the relay, as follows:(11)$$\begin{array}{l} \underset{i\in R,a,b_{k,1},b_{k,2}}{\max}\theta_{i},\\ s.t.\;{\left|b_{k,1}\right|}^{2}+{\left|b_{k,2}\right|}^{2}\leq P_{max},\;k\in N_{r},\\ \;\;\;\;\;\;\;\;b_{k,1}=0,\;{\left|b_{k,2}\right|}^{2}\leq P_{max},\;k\in {\bar{N}}_{r},\\ \;\;\;\;\;\;\;\;P_{r,\mathrm{TX}}\leq P_{max}. \end{array}$$

Relay selection is divided into two steps. The remaining energy per node is not considered when selecting candidate relay nodes, but is considered when selecting one as a relay from the candidate relay nodes. This is due to the following reason: in slow channel fading, the channel gain is stable for a relatively long time, but the remaining energy per node changes with time. Then, it is possible to compute the candidate relay nodes at a long interval, while periodically selecting a relay at a short interval, so as to distribute the load and power consumption to different nodes.

#### 3.3.2. Problem of the SLAM

We performed some simulations to evaluate the impact of parameter α, using the condition in Table 1. Figure 5 shows how network lifetime and MSE change with α. With a large α, the metric focuses more on remaining energy instead of MSE. Therefore, MSE increases with α. However, the lifetime does not always increase with α. It decreases when α becomes greater than 0.6, which is different from the expectation.

During the simulation process, we found that when a node in the network was far away from both the relay node and the sink, the channel gain of this node was very low, causing it to use high transmission power in the communication. As a result, among the nodes in the network, the first node that runs out of energy is not the node that acts as the relay node. The remaining energy of the relay node is not used to improve the network lifetime in this case.

### 3.4. Extending the Basic Method (SLAM-E)

To solve the problem of the basic method, we designed Algorithm 2 to explicitly predict the remaining lifetime of the network when a relay node is selected.

In Algorithm 2, regarding each candidate in R as a relay in turn, we predicted the network lifetime based on the remaining energy and current power consumption of each node (lines 5 to 9). Here, network lifetime is the minimum of all nodes’ lifetime (line 10). By iterating over all the candidate relay nodes in turn, the network lifetimes corresponding to all the candidate relay nodes are obtained.

Then, we updated the metric in Equation (10), replacing $w_{E,i}$ by $w_{L,i}$, as follows.
(12)$$w_{L,i}=\frac{L_{i}-\underset{i\in R}{\min}L_{i}}{\underset{i\in R}{\max}L_{i}-\underset{i\in R}{\min}L_{i}}.$$

The other process is the same as the basic method in Algorithm 1.
**Algorithm 2:** Predict the remaining lifetime of the network**Procedure**: Find network lifetime**Input**: K (Number of nodes), R, E (the set of all nodes’ remaining battery);**Output**: L recording the lifetime of each node in R;**For** i∈R **do**|  Regard node i as relay r, and get remaining energy of all nodes ($E_{j},j=1,\cdots ,K$);|  **For** j = 1, 2, …, K, j≠i
**do**|  |  Calculate the transmission power of node *j* ($P_{j}^{i}$);|  |  $Lifetime_{j}^{i}=E_{j}/P_{j}^{i}$;|  **End**
|  Network lifetime computation: $L_{i}=\underset{j}{\min}Lifetime_{j}^{i}$;**End****Return** *L*;

### 3.5. Analysis of Computational Complexity

In the network, the number of nodes is K, and the number of candidate relay nodes is N=|R|. Here, |R| denotes the number of the candidate relay nodes. Both the basic and extended methods use Algorithm 1 to compute the set of candidate relay nodes. In Algorithm 1, a loop with a loop count K is performed from line 4 to line 15, and another inner loop with a loop count K−1 is performed from line 9 to line 13. In line 11, the CRP algorithm is used, whose time complexity is $T\left(\cdot \right)=NK^{2}\log K$. Therefore, the time complexity of Algorithm 1 is$\;T\left(\cdot \right)=NK^{2}\log K\cdot K\left(K-1\right)=\;O\left(NK^{4}\log K\right)$.

In Algorithm 2, an outer loop with a loop count N=|R| is performed from line 4 to line 11, and an inner loop with a loop count K−1 is performed from line 6 to line 9. Therefore, the time complexity of Algorithm 2 is$\;T\left(\cdot \right)=N\cdot K\;\leq \;O\left(K^{2}\right)$.

The basic method (SLAM) directly computes the evaluation metric by Equations (10)–(11), and the extra computation is T(·)=N= O(N). The extended method (SLAM-E) needs to predict the network lifetime, and the extra computation is T(·)=N·K+N= O(NK).

Then, the overall computation complexity of both methods is $O\left(NK^{4}\log K\right)$, decided by Algorithm 1.

## 4. Simulation Results

The proposed method was verified by simulation, using “MATLAB” (provided by MathWorks Inc.). Network lifetime and MSE are used as the main evaluation metrics. The basic AirComp [10] is also included in the evaluation for a comparison.

### 4.1. Simulation Parameters

Main simulation parameters are shown in Table 1. 

The channel gain between two nodes is mainly determined by their distance. Here, we use the free-space model and the two-ray ground-reflection model to simulate the path loss. The threshold for the signal transmission distance *d* is
(13)$$d=\frac{4\pi h_{t}h_{r}}{\lambda },$$
where $h_{t}$ and $h_{r}$ are the heights of the transmitting and receiving antennas, respectively. λ is the wavelength of the transmitted wave. When the transmission distance is less than *d*, the free-space model is used, otherwise the two-ray ground-reflection model is used. In addition, block Rayleigh fading is assumed, and the channel is stable within two time slots.

In the evaluation, we considered an area of 200 m by 200 m, and the sink was located at the center of the area (100, 100). In order to evaluate the scalability of the method, the number of nodes was set to 20, 30, and 40, respectively, and nodes were randomly distributed in the experiment area, as shown in Figure 6. On average, the channel gain may be strong enough. However, for nodes at the cell edge or in deep fading, their channel gains will be poor, which requires the use of a large transmission power to reduce the MSE. In such cases, using relay helps to reduce node transmission power and extend network lifetime.

Based on the above conditions, we carried out the simulation experiments 1000 times, and the average results are reported.

### 4.2. Impact of Parameter α

First, we evaluated the impact of α on network lifetime and MSE, and the results are shown in Figure 7. In both SLAM and SLAM-E, MSE increases with α. As for the lifetime, in the SLAM method, the lifetime, increasing at first, starts to decrease after α becomes greater than 0.6, which is already explained in Figure 5. In comparison, the lifetime in SLAM-E continuously increases with α, indicating that SLAM-E solves the problem of SLAM and effectively prolongs the network lifetime. Hereafter, α=0.5 is chosen for SLAM and SLAM-E unless specified otherwise.

### 4.3. Result of Lifetime

Next, we evaluated the network lifetime using six settings. Three of these used the SLAM method and the others used the SLAM-E method. In the same method, the number of nodes was set to 20, 30, and 40, respectively. All of them adopt the CRP method when relaying signals. 

Figure 8 shows the CDF (cumulative distribution function) of the network lifetime for six settings. The CDF results confirm that SLAM-E prolongs the network lifetime compared with SLAM, under the same number of nodes and the same value of α. Additionally, the lifetime increases with the number of nodes in both SLAM and SLAM-E.

Figure 9 shows the average network lifetime of four methods. SLAM-CRP improves network lifetime by 28.1% compared with the basic AirComp method, and SLAM-E-CRP further improves the network lifetime by 15.9% compared with SLAM-CRP. Surprisingly, using CRP together with AirComp leads to a worse network lifetime. This is because it focuses more on reducing MSE and nodes consume more power than in AirComp.

### 4.4. Result of MSE

Figure 10 shows the CDF of MSE for nine settings. It is clear that MSE increases with the number of nodes. This is straightforward because it becomes more difficult to align signals when there are more nodes. SLAM-E-CRP reduces MSE a little compared with SLAM-CRP, under the same number of nodes.

Figure 11 shows the average MSE of the four methods. The basic AirComp has the worst MSE. In comparison, the other 3 methods using relay greatly reduce MSE. When the number of nodes is 40, SLAM-CRP reduces the MSE by 45.2% compared with AirComp, and SLAM-E-CRP further reduces the MSE by 2% compared with SLAM-CRP.

The average lifetime and MSE are summarized in Table 2. The MSE of AirComp-CRP is a little less than that of SLAM-CRP and SLAM-E-CRP. This is because AirComp-CRP focuses more on reducing MSE, at the cost of more transmission power (shorter network lifetime). In comparison, SLAM-CRP and its enhancement, SLAM-E-CRP, achieve a better tradeoff between lifetime and MSE, which helps to better exploit the relay node to improve the overall performance.

Initially, we thought that the relay node was the bottleneck because its power consumption increases with the number of signals using relay. On this basis, we designed SLAM. However, in the evaluation, we found that some nodes may still use the maximal transmission power even when using the relay node and deplete their energy faster than the relay node because the relay node is dynamically switched. SLAM-E helps to solve this problem, by directly predicting network lifetime. The experiment results confirm that (i) SLAM far outperforms the basic over-the-air computation model, and (ii) SLAM-E further extends the network lifetime while maintaining a similar MSE to that of the SLAM method.

## 5. Conclusions

In this paper, in order to improve the lifetime of AirComp and reduce MSE, we have proposed a new method (SLAM) to determine the candidate relay nodes and select the optimal relay node. The set of candidate relay nodes is mainly composed of nodes with high channel gains and a lot of remaining energy that can serve as the relay node to help other nodes with low channel gains. A new metric was designed to consider both MSE and lifetime in the selection of the relay node. We further enhanced the basic method to explicitly predict network lifetime in the relay selection, which helps to prolong the network lifetime. Simulation results confirm that the proposed method achieves a better tradeoff between network lifetime and MSE compared with previous methods. In the future, we will investigate the impact of mobile sink, and try to use multiple relay nodes simultaneously to further improve the performance.

## Figures and Tables

**Figure 1 sensors-23-03824-f001:**
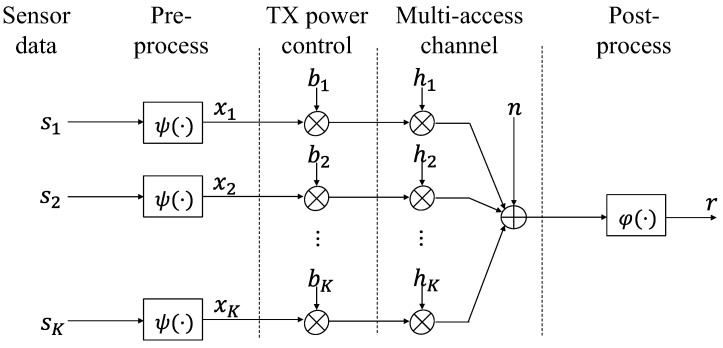
Communication model of AirComp.

**Figure 2 sensors-23-03824-f002:**
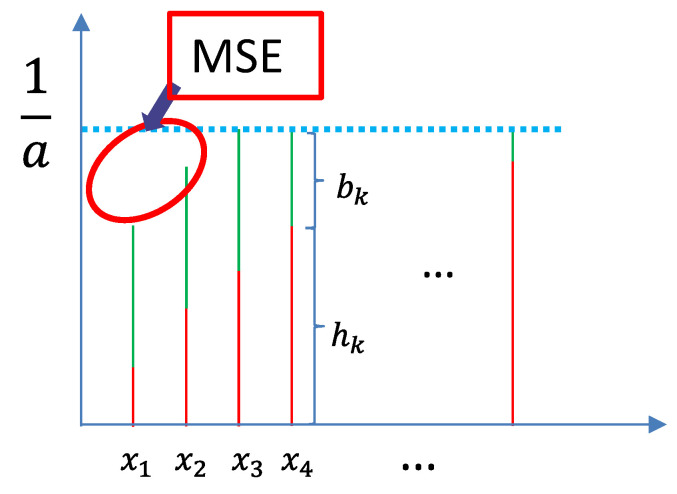
MSE in the signal caused by low channel gain (the length of a line represents the absolute value of a variable).

**Figure 3 sensors-23-03824-f003:**
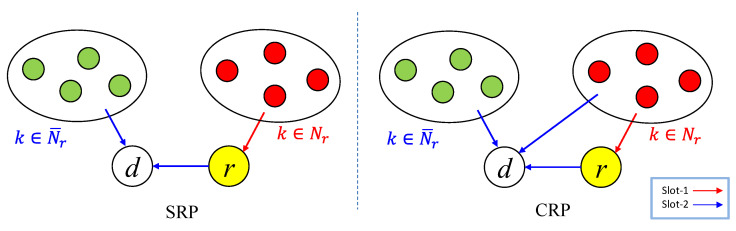
Relay communication policy of SRP and CRP (A circle represents a group of nodes).

**Figure 4 sensors-23-03824-f004:**
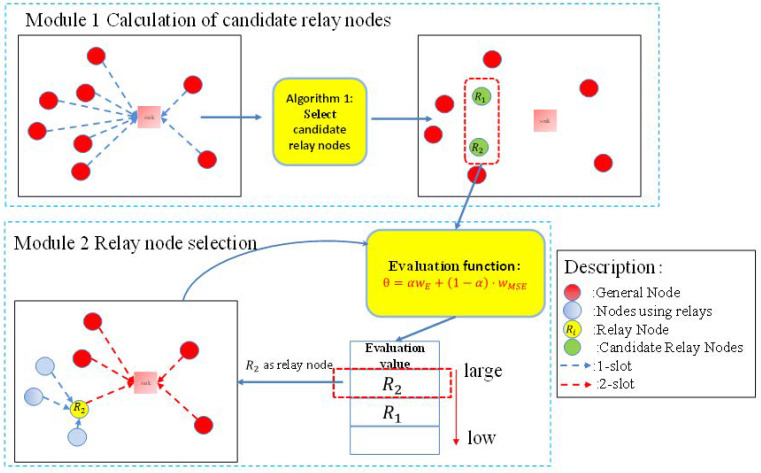
System model of SLAM.

**Figure 5 sensors-23-03824-f005:**
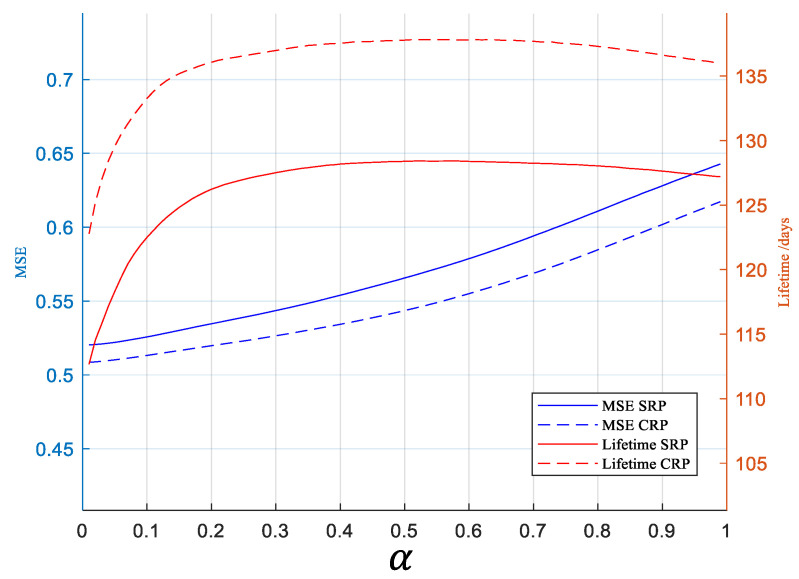
Variation of MSE and lifetime with respect to the parameter α in Equation (10).

**Figure 6 sensors-23-03824-f006:**
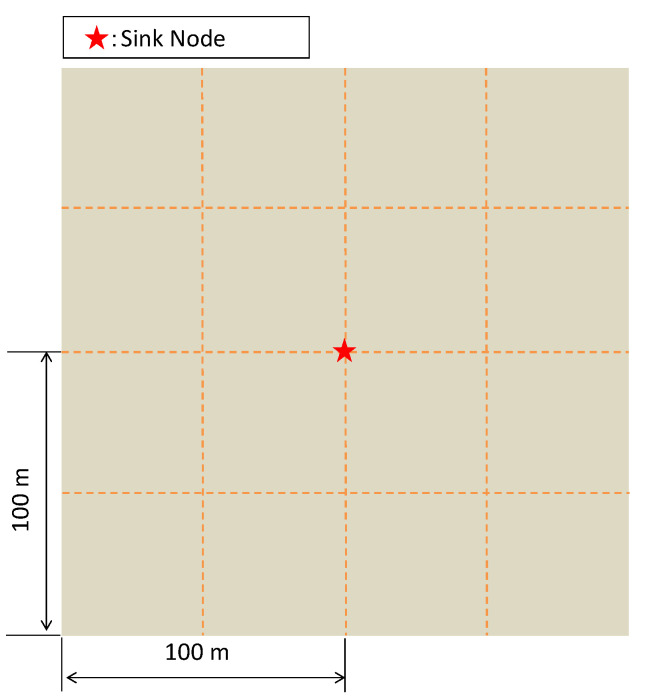
Square area for deploying sensor nodes. The sink is located at the center.

**Figure 7 sensors-23-03824-f007:**
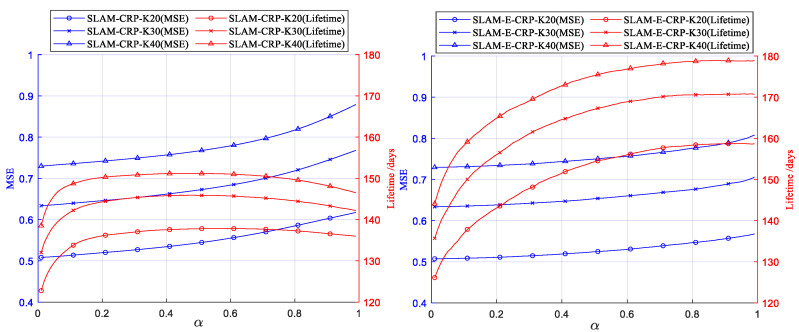
Variation of network lifetime and MSE with respect to α in Equation (10) (SLAM and SLAM-E).

**Figure 8 sensors-23-03824-f008:**
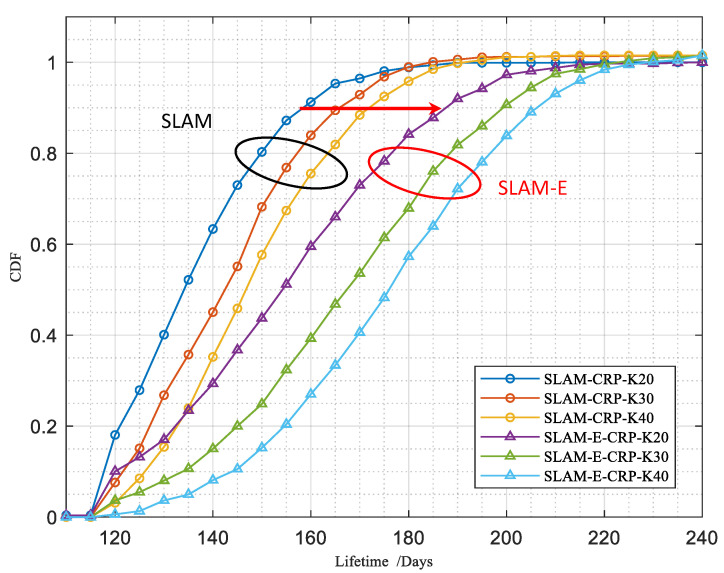
Cumulative distribution of network lifetime (α=0.5, a large circle is used to group results of the same method together).

**Figure 9 sensors-23-03824-f009:**
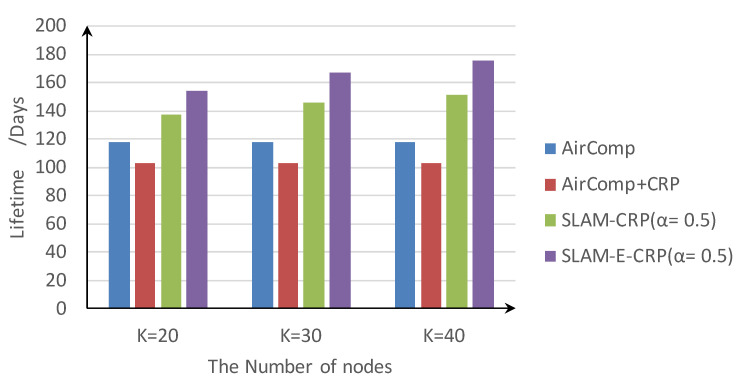
Variation of network lifetime with the number of nodes (α=0.5).

**Figure 10 sensors-23-03824-f010:**
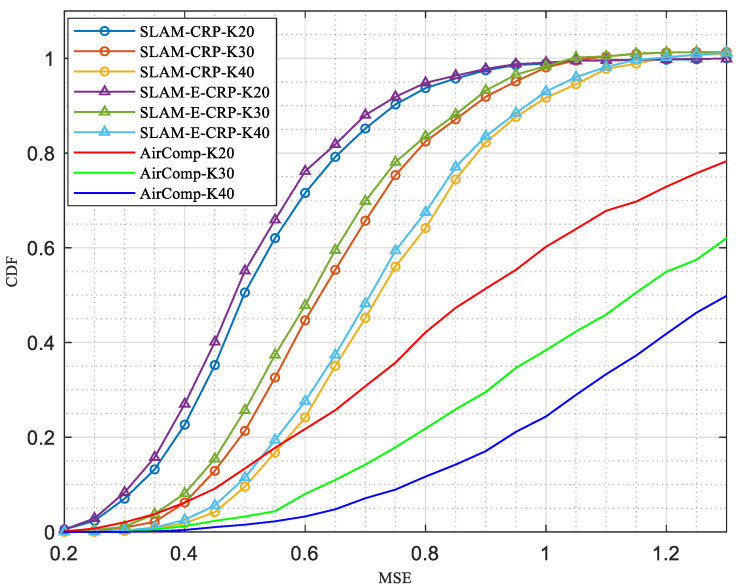
Cumulative distribution of MSE (α=0.5).

**Figure 11 sensors-23-03824-f011:**
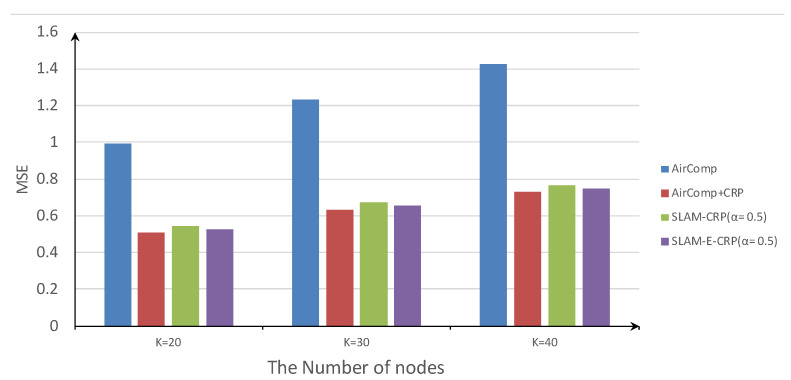
Average MSE of four methods.

**Table 1 sensors-23-03824-t001:** Main notations and their default values.

Parameters	Descriptions	Default Values
*K*	Number of nodes	20
*H*	Height of antenna	1.5 m
$h_{gain}$	Antenna gain	1
$h_{loss}$	Antenna loss	1.5
*Range*	Network range	200 m × 200 m
$Sink_{pos}$	Position of sink node	(100, 100)
$P_{max}$	Maximum limit power	10 dBm
$\sigma^{2}$	Nosie power	1
*J*	Battery capacity	10,000 J
$P_{idle}$	Idle power of node	0.16 mW
$T_{trans}$	Time of transmit	5 s/min
$T_{idle}$	Time of idle	55 s/min
*f*	Frequency of analog wave	2.4 GHz

**Table 2 sensors-23-03824-t002:** Average lifetime and MSE of four methods (*K* = 40, α = 0.5).

Method	Average Lifetime (Day)	Average MSE
AirComp [10]	118	1.424
AirComp-CRP [15]	103.4	0.729
SLAM-CRP	151.2	0.766
SLAM-E-CRP	175.2	0.750

## Data Availability

Not applicable.

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
