# Peer review of "Relay Selection for Over-the-Air Computation Achieving Both Long Lifetime and High Reliability"

_sensors, 2023, doi:10.3390/s23083824_

Round 1

Reviewer 1 Report

This paper studies the best relay selection for the over-the-air computation in wireless sensor networks. The reviewer has some concerns given as follows:

(1) The related works need to be surveyed more sufficiently to illustate the position of this work in this area.

(2) It seems the AF protocol is used for the relaying, then the aggragate noise will affect the performance of the receiver. Why not also consider the DF protocol?

(3) The selection metric of the best relay should be clarified as well as the selection procedure should be clarified. 

(4) The impacts of various parameters, such as the relay locations, the distributions of sensors, the transmit powers of relay, etc should be studied.

(5) The definition of network lifetime should be clarified, as it is a main metric of this work. 

Author Response

Q1: The related works need to be surveyed more sufficiently to illustrate the position of this work in this area.

A1: Thanks for this comment. We have added several latest references and revised the introduction part to better illustrate the position of this work.

Q2: It seems the AF protocol is used for the relaying, then the aggregate noise will affect the performance of the receiver. Why not also consider the DF protocol?

A2: Thank you for this comment. As mentioned in the 2nd paragraph of Sec.1, by using analog AirComp, the computation error can be made smaller than that of digital schemes when using the same amount of resources (power & bandwidth) [5]. Therefore, in this paper, the AF-based relay protocol is used. We have also revised the 2nd paragraph of Sec.3.1 to clarify this.

Q3: The selection metric of the best relay should be clarified as well as the selection procedure should be clarified. 

A: Thank you for this suggestion. After selecting candidate relay nodes by Algorithm 1, for each candidate relay node, we compute its metric by (10). Then, the node with the largest metric is selected as the relay node. We have re-written Sec. 3.3.1 to better explain this.

Q4: The impacts of various parameters, such as the relay locations, the distributions of sensors, the transmit powers of relay, etc. should be studied.

A: Thank you for this comment. First, in the evaluation, all sensors are randomly distributed in the experiment area. The distribution changes per seed, and we did the experiment with 1000 seeds (1000 networks) to calculate the statistics of lifetime and MSE.

Different from [15], we do not use a dedicated relay, instead a sensor node is selected as a relay, which changes periodically. As a result, the relay position (the position of the sensor selected as the relay) changes dynamically. The transmission power of the relay node is decided by the nodes using the relay, under the maximal power constraint. Therefore, in the evaluation, the transmission power of the relay node always approaches but is less than the maximal power constraint. We clarified this below Eq. (9).

Q5: The definition of network lifetime should be clarified, as it is a main metric of this work. 

A5: Thank you for this suggestion. We have added the definition of lifetime at the beginning of Sec. 3.

Reviewer 2 Report

The paper is interesting and opens the door to use this approach to optimize lifetime and computation in WSN, with the goal of basic monitoring process, if only statistical values are needed, by using over the air computation (AirComp as the authors abbreviate. However, sometimes nodes require high power transmission. Thus, in this paper, the authors show a joint solution for relaying selection, considering both computation error and power consumption. In addition, they also consider lifetime. The results are based on simulation. Therefore, they call this method as relay Selection considering both Lifetime And MSE (SLAM).

Although the wording is clear, the authors can improve this paper by taking into account the following comments.

>>List of comments and suggestions:

In general, the authors should focus on their design and try to explain it better, trying to consider real scenarios and or similar approaches.

Review sentences: “If only statistical values of the data are needed, using Over-the-Air Computation (AirComp) can efficiently realize data collection and computation.” Check this sentence, since realize could have a wrong meaning. I suggest: “realize” -> perform or carry out

In line 93, it is said: “We first calculate a set of candidate relay nodes based on channel gains.” How is this done (estimate the channel gain) in a WSN node? It requires further explanation.

In line 113, it is said: “These signals are summed up at the sink antenna, which is postprocessed by Ï•(). This is also related to collaborative beamforming. In this scenario, synchronization with low cost nodes is very complex. These things should be justified or explained, or just define assumptions properly. For instance, In the introduction or related research section, it should be nice to consider and include some other references dealing with the background problem to cooperative communication such as the one described in “feasibility of a stochastic collaborative beamforming for long range communications in wireless sensor networks” with digital identifier 10.3390/electronics7120417 as an reference to review the underlying problems.

In line 121, it is said: In the same paragraph, “Because it is possible to adjust the phase” … It should be nice if the authors could add some more detail?

Besides, in this context, it should be interesting to include some other references to motivate alternatives as path diversity related to different alternatives for relaying techniques such as the one shown in “ "Path diversity gain with network coding and multipath transmission in wireless mesh networks," 2010 IEEE International Symposium on "A World of Wireless, Mobile and Multimedia Networks" (WoWMoM), 2010.

 In Figure 6, in the caption “ Variation of MSE and lifetime with respect to the parameter ?” it should be better to remind what is  ? and mention the equation where it is used, for easy-reading.

Also, the notation in Figure 8. Variation of network lifetime and MSE with respect to ? (SLAM and SLAM-E). We cannot read it properly.

A minor issue is, in the equations 1-9, it should be better to avoid using an extra “.” point at the end of each expression.

In Algorithm 1, line 10 “Assume top j nodes in D use the relay node;” , it should  D -> D_k

In line 219 , section 3.2 it is said: “In Algorithm 1, we assume each ordinary node in turn as a relay node.” It should be better to justify this assumption. Why is not considered the remaining energy? Later, in section 3.3 it is considered. It seems a bit confusing. The authors could consider a slightly better wording.

In addition, in Table 1 check second line “H Height of antenna 2.4GHz”. I guess this is a typo. Also, based on the detail of Table 1, for the simulations using a square of 200m x 200m it seems that relaying is not necessary, although it would depend on the technology used. Could the authors discuss this?

I guess that if Algorithm 1 is in bold, Algorithm 2 should be in bold too in the paper.

In section 3.5 when discussing the “Analysis of Computational Complexity”. Why is not considered the complexity of Algorithm 2? It should be better to add an explanation.

It should be interesting to highlight the difference between SLAM and SLAM-E, since SLAM also considers the lifetime. Also, I guess that the authors are dealing with two different things, on one hand the reaming energy and in the other the life time. Please, could you explain and clarify this?

In Algorithm 2, how is calculated the power consumption of node j ? is something measured locally at each node? How? Is this feasible in a real deployment=

In section 4.3 it is said in line 313, “All of them adopt the CRP method when relaying signals.” Why SRP is not considered? Although it is mentioned in lines 195 and 196, “it is already proven that CRP works better than SRP in the amplify-and-forward based relay [15]. Therefore, in the following we mainly study how to dynamically select a relay for CRP.”. In my opinion, it is required a better justification in this paper.

Justify why in Figure 10 it is used “alfa=0.5” in SLAM-CRP(α= 0.5) SLAM-E-CRP(α= 0.5).

Finally, the conclusions should summarize the figures achieved and explained in the results.

Author Response

Q1: In general, the authors should focus on their design and try to explain it better, trying to consider real scenarios and or similar approaches.

A1: Thank you for this comment. In the experiment, we consider the random distribution of sensor nodes in the network, and potential fading in the channel, which is close to the real environment.

Q2: Review sentences: “If only statistical values of the data are needed, using Over-the-Air Computation (AirComp) can efficiently realize data collection and computation.” Check this sentence, since realize could have a wrong meaning. I suggest: “realize” -> perform or carry out

A2: Thank you for this comment. Following this comment, we have replaced the word “realize” by “perform” in the abstract.

Q3: In line 93, it is said: “We first calculate a set of candidate relay nodes based on channel gains.” How is this done (estimate the channel gain) in a WSN node? It requires further explanation.

A3: The estimation of channel gain is not the focus of this paper. In Section 3.1 we have assumed that the sink knows the channel coefficients () between itself and all nodes and the channel coefficients between all nodes.

Q4: In line 113, it is said: “These signals are summed up at the sink antenna, which is postprocessed by Ï•(⋅). “ This is also related to collaborative beamforming. In this scenario, synchronization with low cost nodes is very complex. These things should be justified or explained, or just define assumptions properly. For instance, In the introduction or related research section, it should be nice to consider and include some other references dealing with the background problem to cooperative communication such as the one described in “feasibility of a stochastic collaborative beamforming for long range communications in wireless sensor networks” with digital identifier 10.3390/electronics7120417 as an reference to review the underlying problems.

A: Thank you for this comment. In this article, we assume that synchronization can be achieved in Sec.3.1. The alignment of signals in AirComp is different from that in beamforming. We have clarified this in Sec. 1, and also included the above reference.

Q5: In line 121, it is said: In the same paragraph, “Because it is possible to adjust the phase” … It should be nice if the authors could add some more detail? Besides, in this context, it should be interesting to include some other references to motivate alternatives as path diversity related to different alternatives for relaying techniques such as the one shown in “ "Path diversity gain with network coding and multipath transmission in wireless mesh networks," 2010 IEEE International Symposium on "A World of Wireless, Mobile and Multimedia Networks" (WoWMoM), 2010.

A5: Thanks for this comment. ,  and  usually are complex numbers. Because the phase of  can be freely adjusted so that the product, , is always positive and real, in the analysis, we assume that  and  are positive real, for simplicity. This is the same as in previous work [10]. In the evaluation, complex numbers are used instead. We also included the mentioned reference.

Q6: In Figure 6, in the caption “ Variation of MSE and lifetime with respect to the parameter ?” it should be better to remind what is ? and mention the equation where it is used, for easy-reading. Also, the notation in Figure 8. Variation of network lifetime and MSE with respect to ? (SLAM and SLAM-E). We cannot read it properly.

A6: Thank you for this comment. We have modified the caption of Figure 5,7, to explain where ? appears. We also used larger font for the legend in Fig. 7.

Q7: A minor issue is, in the equations 1-9, it should be better to avoid using an extra “.” point at the end of each expression.

A7: Because equations are a part of the text, we prefer to add a “,” or “.” to show the end of the text.

Q8: In Algorithm 1, line 10 “Assume top j nodes in D use the relay node;” , it should  D -> D_k. And I guess that if Algorithm 1 is in bold, Algorithm 2 should be in bold too in the paper.

A: Thank you for this comment. We have refined the definition of  in line 7 of Algorithm 1. Now, Algorithm 2 also appears in bold font.

Q9: In line 219, section 3.2 it is said: “In Algorithm 1, we assume each ordinary node in turn as a relay node.” It should be better to justify this assumption. Why is not considered the remaining energy? Later, in section 3.3 it is considered. It seems a bit confusing. The authors could consider a slightly better wording.

A9: In Sec.3.2, we select candidate relay nodes by Algorithm 1, and then in Sec. 3.3, we select one candidate node as the relay according to the evaluation metric. “Assume each ordinary node in turn as a relay” in Algorithm 1 only means we regard each node in turn as a potential relay, and compute the corresponding MSE and lifetime. We use “assume” only to distinguish the words “select” where the relay node is actually selected. To avoid confusion, we change this word to “regard”.

Remaining energy is not considered when selecting candidate relay nodes, but considered when selecting the relay from candidate relay nodes. This is due to the following reason. In slow channel fading, the channel gain is stable for a relatively long time, but the remaining energy changes with time. Then, it is possible to compute the candidate relay nodes at a long interval, while selecting a relay at a short interval. We have clarified this in Sec. 3.3.1.

Q10: In addition, in Table 1 check second line “H Height of antenna 2.4GHz”. I guess this is a typo. Also, based on the detail of Table 1, for the simulations using a square of 200m x 200m it seems that relaying is not necessary, although it would depend on the technology used. Could the authors discuss this?

A10: Thank you for pointing out the typo. We have changed “H Height of antenna 2.4GHz” to “H Height of antenna 1.5 m”. In a square area of 200mx200m, on average the channel gain may be strong enough, but for nodes at cell edge or in deep fading, their channel gains will be poor, which requires to use large transmission power to reduce the computation error. Using relay helps to reduce node transmission power and extend network lifetime. We have revised Sec. 4.1 to clarify this.

Q11: In section 3.5 when discussing the “Analysis of Computational Complexity”. Why is not considered the complexity of Algorithm 2? It should be better to add an explanation.

A11: Originally, the complexity of Algorithm 2 was not analyzed because we thought that it is relatively simple. To avoid unnecessary misunderstanding, we have revised this part of the article and added a complexity analysis for Algorithm 2 as well.

Q12: It should be interesting to highlight the difference between SLAM and SLAM-E, since SLAM also considers the lifetime. Also, I guess that the authors are dealing with two different things, on one hand the remaining energy and in the other the life time. Please, could you explain and clarify this?

A12: Thanks for this comment. Initially, we thought that the relay node is the bottleneck because its power consumption increases with the number of signals using relay. On this basis, we designed SLAM. But when evaluating the SLAM method, we found that when the parameter ? in the evaluation metric increases (? >0.6), the lifetime of the network does not increase as expected. Upon checking the remaining energy of all nodes, we found that the energy of some nodes was already exhausted while there is still remaining energy at the relay node. To solve this problem, we designed Algorithm 2 and directly considered network lifetime in the evaluation metric, in case that some far nodes will deplete their battery before the relay node does. Finally, we achieved the result as expected. To summarize, SLAM-E fully utilizes the remaining energy of the candidate relay nodes in the network to help far nodes, and further extends network lifetime compared with SLAM.

Q13: In Algorithm 2, how is calculated the power consumption of node j? is something measured locally at each node? How? Is this feasible in a real deployment=

A13: At the sink, with the channel gains to all nodes known, the sink node can compute the transmission power of node , which is  for direct transmission and  for relay transmission, as shown in Eq.(11).

Q14: In section 4.3 it is said in line 313, “All of them adopt the CRP method when relaying signals.” Why SRP is not considered? Although it is mentioned in lines 195 and 196, “it is already proven that CRP works better than SRP in the amplify-and-forward based relay [15]. Therefore, in the following we mainly study how to dynamically select a relay for CRP.”. In my opinion, it is required a better justification in this paper.

A14: The experimental comparison of CRP and SRP has actually been done, but we did not explain it in detail in this paper because we felt that it was unnecessary to compare them again due to the space and the fact that we had already confirmed it in paper [15].

Q15: Justify why in Figure 10 it is used “alfa=0.5” in SLAM-CRP(α= 0.5) SLAM-E-CRP(α= 0.5).

A15: Based on the results in Figure 7, the lifetime in SLAM-CRP reaches the peak at around  where MSE is still relatively low. Therefore,  is chosen for SLAM-CRP. For a fair comparison,  is also chosen for SLAM-E-CRP.

We have added one sentence in the 1st paragraph of Sec. 4.2 to clarify this.

Q16: Finally, the conclusions should summarize the figures achieved and explained in the results.

A16: Thanks for this suggestion. We have added one paragraph just before the conclusion, to summarize all the results.

Round 2

Reviewer 1 Report

The reviewer has no further comments. 

Reviewer 2 Report

The paper has been improved and it is worth for publication.